# Boredom Makes Me Sick: Adolescents’ Boredom Trajectories and Their Health-Related Quality of Life

**DOI:** 10.3390/ijerph18126308

**Published:** 2021-06-10

**Authors:** Manuel M. Schwartze, Anne C. Frenzel, Thomas Goetz, Reinhard Pekrun, Corinna Reck, Anton K.G. Marx, Daniel Fiedler

**Affiliations:** 1Department of Psychology, Ludwig-Maximilians-Universität München, 80539 Munich, Germany; frenzel@psy.lmu.de (A.C.F.); pekrun@lmu.de (R.P.); corinna.reck@psy.lmu.de (C.R.); anton.marx@psy.lmu.de (A.K.G.M.); daniel.fiedler@psy.lmu.de (D.F.); 2Department of Developmental and Educational Psychology, Faculty of Psychology, University of Vienna, 1010 Vienna, Austria; thomas.goetz@univie.ac.at; 3Department of Psychology, University of Essex, Colchester CO4 3SQ, UK; 4Institute for Positive Psychology and Education, Australian Catholic University, North Sydney, NSW 2060, Australia

**Keywords:** achievement emotions, boredom, adolescents, health-related quality of life

## Abstract

Existing research shows consistent links between boredom and depression, somatic complaints, substance abuse, or obesity and eating disorders. However, comparatively little is known about potential psychological and physical health-related correlates of academic boredom. Evidence for such a relationship can be derived from the literature, as boredom has adverse consequences in both work and achievement-related settings. The present study investigates latent correlations of 1.484 adolescents’ (*M*_age_ = 13.23) mathematics boredom scores at three time points during a semester in 2018/19 and their Rasch scaled health-related quality of life (HRQoL). Moreover, we applied latent growth curve modeling to estimate boredom trajectories across the semester and determined the relationship between the latent growth parameters of student boredom and HRQoL in bivariate correlation analyses. Our results show that boredom is significantly negatively linked with all HRQoL dimensions (physical well-being, psychological well-being, autonomy and parent relation, social support and peers, school environment [SCH], and general HRQoL [GH]). Furthermore, stronger increases in boredom across the semester were negatively associated with SCH scores and GH. In conclusion, given that boredom is negatively linked with HRQoL and that stronger boredom growth is linked with more severe health-related problems, signs of academic boredom could be an early warning signal for adolescents’ potentially severe problems.

## 1. Introduction

While there are several different definitions of boredom, most of them agree that boredom experiences are typically characterized by a certain degree of negative valence, coupled with attentional issues, the perception of time passing slowly, and insufficient and dissatisfactory stimulation, challenge and meaning [1]. Boredom is one of the most commonly experienced emotions in educational settings [2,3]. Particularly during the adolescent years, students report elevated levels of academic boredom [4], and among U.S. adolescents, the overall experience of boredom increased steadily from 2008 to 2017 [5]. Some philosophical notions of boredom emphasized its benefits [6,7] and also in the psychological literature, it has been argued that boredom can be considered functional, for example, in the context of willpower [8]. However, academic boredom in particular, has been shown to be a largely adverse emotional experience [6] and there is consistent empirical evidence that academic boredom is linked with a multitude of problematic academic outcomes, including higher levels of achievement-related anger, anxiety, and shame [9,10], reduced motivation and effort [11,12,13], lower academic achievement [10,14,15,16,17], and dropping out of school [18,19].

Similarly, in the context of work, there is evidence that boring, monotonous working conditions are associated with emotional health complaints [20] and physical health problems, such as visual and musculoskeletal complaints, asthma, bronchitis, and hand tremors [20,21], as well as stress-related health issues such as cardiovascular diseases [22]. Furthermore, there is ample empirical evidence that general boredom proneness and leisure boredom are linked with depression [23,24,25,26,27], somatic complaints [26], substance abuse [28,29,30,31], obesity, eating disorders [32,33], and borderline personality disorder diagnosis [34,35].

However, for educational settings, evidence on psychological and physical health correlates of boredom is largely lacking. Hypotheses on such correlates can be derived from the above-mentioned literature as boredom is likely to have similar adverse consequences in work and education. Within this literature, it is discussed that boredom may exacerbate health problems, such as increased food consumption in obese individuals [32]. Conversely, it can also be the case that unhealthy behaviors such as substance abuse intensify experiences of boredom [28]. It is therefore conceivable that boredom and health are linked through reciprocal effects. Furthermore, the boredom cascade model suggests that boredom results in frustration and maladaptive escape behaviors (such as impulsive behavior induced by identity disturbance) that fuel chronic feelings of emptiness, which, in turn, generate boredom [35].

Given the lack of studies on health correlates of boredom in education, we aimed to explore whether academic boredom is linked with general psychological and physical health problems. If academic boredom is linked with psychological and physical health-related variables, then students’ boredom experienced in school could be interpreted as an early warning signal for potentially severe health problems in adolescence. With over 1.3 billion primary and secondary school students worldwide [36], schools play a crucial role in determining not only educational outcomes but also health [37], thus making them primary agents to protect and improve public health.

Boredom in school has been shown to be highly domain-specific [38]. Many recent studies on academic boredom focused on the subject of mathematics, as this subject takes an outstanding role in modern societies. Scholastic success in mathematics is important for a wide range of professions [39] and a predictor of participation in secondary education and expected future salary [40]. For example, research on mathematics boredom addressed the antecedents of boredom, such as control-value appraisals [12,41], links between boredom and achievement [10], and behavioral and personality correlates among highly bored low and high achievers [42]. In line with these studies, we operationalize academic boredom in a domain-specific way, focusing on the domain of mathematics. We conceptualized mathematics boredom as an individual differences construct [43], proposing that individuals systematically differ in their tendency to experience boredom in mathematics. We asked secondary school students about their habitual tendencies to experience mathematics boredom three times across a semester, as we aimed to assess a construct that is between trait and state dimensions, albeit more on the trait side.

The key goal of the present longitudinal study was to expand our knowledge on the relevance of mathematics boredom, by investigating whether such domain-specific boredom was linked with students’ health-related quality of life (HRQoL), as measured with the KIDSCREEN at the end of the semester [44]. Here, HRQoL is described as a multidimensional construct covering physical, emotional, mental, social, and behavioral components of well-being, functioning as perceived by the individual [45]. Developed simultaneously in several countries and validated in a large representative sample across 13 European countries, the KIDSCREEN is considered a valid measure of HRQoL and can be used to identify children and adolescents who are at increased risk of developing health problems [46]. We based our first hypothesis on this literature.

**Hypothesis** **1.**
*Trait mathematics boredom is negatively linked with HRQoL.*


In addition to exploring concurrent links of boredom and HRQoL, a second important goal of the present study was to explore if changes in boredom across a semester were linked with HRQoL. Scattered prior research has shown that trajectories of academic boredom show an upward trend over time [47,48,49], with substantial changes between grades 5 and 7 [50], which is also in line with the decline in students’ interest in mathematics during adolescence [51]. We expected to replicate those findings. 

**Hypothesis** **2.**
*Boredom increases across a semester for secondary school students.*


Additionally, prior research has shown that there is variance in adolescents’ boredom growth trajectories. Changes in boredom were linked with contextual and individual factors such as task value, learning style, effort regulation, and academic engagement [49,52]. No study to date seems to have explored whether changes in academic boredom are related to psychological and physical health symptoms. However, based on the existing evidence on concurrent psychological and physiological correlates of boredom, it is conceivable that stronger increases in boredom are linked with lower HRQoL in adolescents, hence our third hypothesis.

**Hypothesis** **3.**
*A greater increase in boredom across a semester is associated with lower subsequent HRQoL.*


## 2. Materials and Methods

### 2.1. Study Design and Procedure

To test our hypotheses that levels and growth trajectories of boredom are linked with HRQoL, we employed data collected in the context of a longitudinal field study in the subject of mathematics. The study included three assessments (T_1_–T_3_) which took place at the beginning of the semester (September 2018, T_1_), in November (T_2_), and in February 2019 (T_3_). Boredom was assessed at all three time points and HRQoL was measured at T_3._ As such, boredom and health outcomes were linked through a prospective design. At T_1_, the questionnaires were handed out in class, filled out at home by the students, and collected again inside sealed envelopes. At T_2_ and T_3_, the data collection was administered by undergraduate research assistants and all questionnaires were filled out in class.

### 2.2. Sample

The overall sample consisted of *N* = 1.484 (*n*_T1_ = 1.400, *n*_T2_ = 1.262, *n*_T3_ = 1.260) secondary school students from 99 classes in 30 schools in Bavaria, Germany. Due to being absent from class, missing consent forms, or a belated decision to participate in the study, 84 participants were missing at T_1_, 222 at T_2_, and 224 at T_3._ Ten participants were missing at both T_1_ and T_2_, 6 at T_1_ and T_3_, and 68 at T_2_ and T_3_. Missing data were handled by using the full information likelihood method (FIML; [53]). At T_1_, students were 9 to 18 years old, with a mean age of 13.23 years (*SD*_age_ = 1.32; 52% girls, *n* = 770; 48% boys, *n* = 714). All tracks of the Bavarian three-track general secondary school system were represented, with 48% (*n* = 708 students) from the upper (Gymnasium), 27% (*n* = 397) from the middle (Realschule), and 26% (*n* = 379) from the lower track (Mittelschule). This distribution across tracks is equivalent with the Bavarian secondary student statistics, with a slight overrepresentation of the Gymnasium student population [54]. The students were in the fifth (*n* = 194), sixth (*n* = 204), seventh (*n* = 613), eighth (*n* = 305), ninth (*n* = 143), and 10th grade (*n* = 25). The majority of the students (86%, *n* = 1.275) were born in Germany, while 26% of them had at least one foreign-born parent (*n*_mother_ = 195, *n*_father_ = 188, *n*_both_ = 256).

### 2.3. Measures

The measures were assessed as part of a more comprehensive self-report survey, assessing behavioral and personality variables.

#### 2.3.1. Boredom

Students’ class-related, habitual (i.e., trait-like) boredom was assessed using the six-item class-related boredom scale of the Achievement Emotions Questionnaire-Mathematics (AEQ-M; [9,55]). For this subscale of the AEQ-M, instructions prompt students to “Please indicate how you feel, typically, during math class.” A sample item is “I am so bored that I can’t stay awake” (see Table A1 in the Appendix B for the full set of items used in this study, in the original German and the English versions). Students responded using a 5-point Likert scale ranging from 1 (*not at all true*) to 5 (*completely true*). The scale showed good internal consistency for all time points, with Cronbach’s α coefficients greater than 0.86 (see Table A2).

#### 2.3.2. Health-Related Quality of Life

The German version of the KIDSCREEN-27 for children and adolescents was used to measure HRQoL [44]. Students were asked “How are you? How do you feel? This is what we would like you to tell us. Please read every question carefully. What answer comes to your mind first? Choose the box that fits your answer best and cross it.** Please try to remember the last week, i.e., the last seven days.” The items comprised of five dimensions: physical well-being (PH, 5 items, e.g., “Have you felt fit and well?”), psychological well-being (PW, 7 items, e.g., “Have you felt sad?”), autonomy and parent relation (PAR, 7 items, e.g., “Have you had enough time for yourself?”), social support and peers (SOC, 4 items, e.g., “Have you been able to rely on your friends?”), and school environment (SCH, 4 items, e.g., “Have you got on well at school?”). Participants responded to all items on a 5-point Likert scale ranging from 1 (*not at all*) to 5 (*extremely*), except for the first item of PH (“In general, how would you say your health is?”) which was scored from 1 (*excellent*) to 5 (*poor*). Negatively worded items were reverse-coded, so that higher scores depict better HRQoL. The scale showed adequate internal consistency for all five dimensions; Cronbach’s α coefficients were all greater than 0.80 (see Table A2). In addition, we also obtained a general HRQoL score from the ten KIDSCREEN-27 items that constitute the KIDSCREEN-10, as suggested by the authors [45]. The development of the KIDSCREEN was based on the probabilistic partial credit model (PCM), from the Rasch family of models [56]. Both the subscale scores and the general HRQoL score have been confirmed to be valid measures of HRQoL across 13 European countries [46]. Accordingly, we also submitted our data to the probabilistic partial credit model, applying an SPSS syntax provided by the KIDSCREEN authors [45]. Next, we translated the obtained Rasch scores into *T*-values, using the norms provided by the KIDSCREEN authors [45]. Comparing our sample with the KIDSCREEN reference population (12 to 18-year-old adolescents), the students in our sample showed average scores within the suggested thresholds for classifying test-values as “normal” or “noticeable” (±½ *SD*—i.e., 5–around the mean T scores of 50) on all scales (see Table A2). As such, our sample can be considered to demonstrate “normal” physical and psychological health on average [45].

### 2.4. Statistical Analysis

We used R 3.6.1 [57], except for the Rasch scale score and norm-related T-score calculation for the KIDSCREEN dimensions for which we used SPSS 26 [58]. The full R code of our analyses is available in the OSF (see data availability statement). The KIDSCREEN score calculation syntaxes can be obtained from the KIDSCREEN authors [45].

Boredom was modeled as a latent construct using the Lavaan package in R [59]. We applied the full information likelihood method (FIML; [53]) to deal with missing data. The MLR estimator (maximum likelihood estimation with robust (Huber–White) standard errors and a scaled test statistic that is (asymptotically) equal to the Yuan–Bentler test statistic), was used to account for non-normal distributions of the data. As a preliminary analysis step, we tested for measurement invariance using the SemTools R package, to make sure that the latent boredom scores were comparable over time [60]. Since χ^2^, and correspondingly also delta χ^2^, are overly sensitive to sample size, we evaluated differences in practical fit indices [61,62,63,64]. We sequentially tested increasingly constrained longitudinal measurement models, namely equivalence of model form (configural), equivalence of factor loadings (metric), and equivalence of item intercepts or thresholds (scalar; [65]). The differences in CFI (−0.002), RMSEA (−0.001), and SRMR (0.001 to 0.004) were clearly below commonly recommended thresholds [66], indicating scalar equivalence. This implies that the latent boredom construct was equally represented by the scale items across the three time points used in the present analyses [67]. In other words, changes in the factor level can be interpreted as reflecting actual differences in the students’ reported experiences of boredom.

To test hypothesis 1, we obtained latent correlations between the boredom scores at T_1_, T_2_, and T_3_ and the HRQoL scores. Concerning hypothesis 2, we applied doubly latent growth curve modeling to estimate boredom trajectories across the semester by using a stepwise confirmatory approach comparing an intercept only (non-growth) with a linear growth model [68]. It is worth noting that our 3-wave-repeated measures design allowed for meaningfully estimating only linear, but not nonlinear (e.g., quadratic) growth [69]. To test hypothesis 3, we determined the relationship between the latent growth parameters of boredom and the HRQoL dimensions, using the growth parameters of boredom (intercept and slope) in bivariate correlation analyses. To account for multiple testing, we adjusted *p*-values using Holm’s method [70], a procedure that has been shown to be more powerful than the original Bonferroni method [71]. 

Students who did not participate at T_3_ (*n* = 224) and therefore, have no HRQoL data did not differ from the overall sample in terms of gender, *t*(294.73) = 0.85, *p*_Holm_ = 1.; age, *t*(298.84) = −2.68, *p*_Holm_ = 0.055; school type, *t*(300.19) = 2.53, *p*_Holm_ = 0.072; math grade, *t*(301.08) = −2.05, *p*_Holm_ = 0.204; country of birth, *t*(256.78) = −0.07, *p*_Holm_ = 1.; boredom T_1_, *t*(287.49) = −0.72, *p*_Holm_ = 1.; and boredom T_2_, *t*(192.37) = −1.04, *p*_Holm_ = 1. As such, missingness at T_3_ was not systematically related with any of those variables.

## 3. Results

Table 1 shows the bivariate correlations between boredom and the HRQoL dimensions. It is worth noting that the correlations between the boredom scores across time were relatively large (*r*s > 0.61), indicating that boredom showed considerable stability over time. Furthermore, confirming hypothesis 1, boredom was significantly negatively linked with all HRQoL dimensions, with the strongest correlations observed for school environment and general HRQoL (*r*s = −0.319 to −0.487). All correlations remained virtually the same when including age and gender as covariates. The correlations imply that boredom was negatively associated with physical well-being (feeling physically exhausted, physically unwell, unfit, having low energy), psychological well-being (having no pleasure in life, feeling depressed, feeling unhappy, having low self-esteem), autonomy and parent relation (feeling restricted, feeling overlooked, not appreciated, feeling finances are restricting life style), social support and peers (feeling excluded, not accepted by peers), school environment (disliking school, negative feelings about school, not doing well at school), and general HRQoL (feeling unhappy, unfit and dissatisfied with regard to family life, peers and school life; [45].

Concerning the growth trajectory of boredom across the semester (hypothesis 2), we used latent growth curve analysis to compare intercept-only (χ^2^ = 860.39, *df* = 137, AIC = 64,693, BIC = 64,968, RMSEA = 0.058) and linear growth models (χ^2^ = 815.53, *df* = 134, AIC = 64,654, BIC = 64,945, RMSEA = 0.057). We settled for the linear growth model as it showed a significantly better fit to the data (χ^2^_diff_ = 38.739, *df*_diff_ = 3, *p* < 0.001). This linear growth model implied that there was a significant, yet small-sized increase in boredom over the three time points (slope = 0.15, 95% CI [0.05, 0.22], *SE* = 0.04, *p* = 0.002, see Figure 1). The variance of this slope also proved to be significantly different from zero (τ^2^ = 0.76, *p* = 0.049). In fact, while the overall latent slope parameter estimate was small and positive, the individual estimates ranged from −2.17 to 2.71. Slope and intercept were negatively correlated (*r* = −0.38, *p* < 0.001).

Having established growth curve parameters of the development of boredom over time, we explored the correlative links between the latent intercept and growth parameters and the HRQoL dimensions next (see Figure A1). In line with the results from the concurrent correlation analyses reported above, the intercept (given our model specification interpretable as students’ levels of boredom at the first measurement T_1_) was significantly negatively correlated with all HRQoL dimensions, indicating that higher boredom levels were linked with lower values on all HRQoL dimensions. Furthermore, and partially in line with hypothesis 3, the slope was significantly linked with the school environment and general HRQoL scores (*r*s = −0.15 and −0.11, respectively). By implication, stronger increases in boredom across the semester were associated with more negative feelings about school and lower general HRQoL (see Table 2). In line with previous studies that showed gender universality of achievement emotion-outcome links [72,73], these relationships proved to be equivalent across genders (see Appendix A).

## 4. Discussion

The present study raised two important questions as to whether levels and change of mathematics boredom across a semester are linked with HRQoL. Confirming our first hypothesis, we provided empirical evidence that high levels of boredom were negatively associated with poorer self-reported physical well-being, psychological well-being, autonomy, parent relations, social support and peer relations, school environment, and general HRQoL, which supports and extends earlier findings on adverse consequences of academic boredom [9,10,11,12,13,15,17,18,19,38,74,75]. Remarkably, context-specifically experienced boredom in mathematics demonstrated substantial negative links with context-transcending health indicators.

Specifically, high boredom in mathematics was substantially associated with lack of pleasure in life, feeling depressed, feeling unhappy, having low self-esteem (PW), disliking school, negative feelings about school, not doing well (SCH), and feeling unhappy, unfit, and generally dissatisfied with regard to family life, peers, and school life (GH). Following Brunswik’s symmetry principles, it is likely that correlations are attenuated if constructs are operationalized at different levels of domain specificity [76]. The pattern of our findings supports the notion of stronger links between contextually closer concepts, as mathematics boredom was most closely linked with the school-related HRQoL dimension (SCH). Furthermore, above and beyond concurrent links, this study also sought to explore the links between trajectories of mathematics boredom and HRQoL. Confirming our second hypothesis and in line with previous research on academic boredom [47,49,50], we observed an increase in boredom during the semester. Boredom intercept and slope proved to be negatively correlated, implying that students who started the schoolyear on higher levels tended to show smaller increases in boredom and vice versa. Confirming hypothesis 3, the dynamics of boredom across the semester were linked with students’ HRQoL. Stronger increases in boredom were linked with more severe health-related problems, specifically with the dimension of school environment (disliking school, negative feelings about school, not doing well), but also with general HRQoL.

Given that our findings are correlational, the mechanisms generating the observed correlations remain open to discussion. On the one hand, boredom and particularly, an increase in boredom across a semester, could be drivers of an adverse health development. It has been shown that boredom during math class occurs when lessons are experienced as either over- or under-challenging and thus, lack meaningful opportunities for engagement [77]. If students repeatedly experience such lessons across the semester and their boredom levels increase over time, they then may withdraw from classes and start to engage in maladaptive escape behaviors. Based on the above-mentioned boredom cascade model, the experience of boredom can then lead to such maladaptive escapes, prompting issues regarding individuals’ mental and physical health [35]. More specifically, repeated and increasingly intense experiences of boredom in mathematics, a subject typically judged as highly important by relevant others (e.g., parents, teachers, society), are likely to increase the use of avoidant and anger-related or acting-out strategies of emotion regulation [50]. Such maladaptive coping and emotion regulation strategies, in turn, are likely to fuel more general psychological and physical health issues, such as vulnerability to infection or cardiovascular disease [78,79].

In contrast to other emotions, where intensity is associated with high value (importance) of the events triggering emotion, boredom is linked to lack of meaning and value—exemplified in the question, “What is it all for?” [6,43,80]. This quest for value expresses an individuals’ lack of purpose or perspective and resembles the core position of meaning in Frankl’s (1959) work on depression and suicide prevention. It is a fundamental human need to want one’s life to be meaningful. Lack of meaning and value, as implied by boredom, thwart this need, thus contributing to health problems [23]. While the emotional experience of boredom has been shown to be psychometrically distinct from depression [23,81,82], the correlative link between boredom and psychological well-being (assessed with items such as “Have you felt sad”) as demonstrated again in our study can be interpreted in that strong experiences of boredom might make students vulnerable to depression.

On the other hand, adverse health conditions could be a driver of boredom experiences in achievement contexts. For example, health conditions such as obesity could lead to a lack of energy, social isolation, and lack of popularity with other students as well as teachers, leading to more boredom in class. Finally, there may also be third variables that can generate both maladaptive levels and trajectories of boredom, and poor health among adolescents, such as extraordinary unfortunate environmental circumstances—the current pandemic-implied school lockdowns being a palpable example, e.g., [83,84]. 

## 5. Limitations

The present study used a robust latent growth curve modeling approach and yielded consistent findings that supported our hypotheses. Nevertheless, the study has limitations that should be considered in interpreting the findings and can be used to derive directions for future research. As noted, the analysis was correlational; future research should replicate the current findings using predictive models and longitudinal designs involving repeated measures of both boredom and health indicators to model their co-development over time. Given that our design only involved three measurement points, we could meaningfully estimate only an intercept-only against a linear growth model, while it is conceivable that the boredom trajectory over time also forms nonlinear trends, e.g., initially strong increases followed by a flattening of the curve; such quadratic trends have, for example, been found for adolescents’ interest loss trajectories during adolescence. Future research could explore corresponding nonlinear growth also for boredom, but a larger number of measurement time points (at least four) would be necessary to meaningfully estimate such nonlinear trend models [69]. Self-report was used to assess both boredom and health problems; future studies should complement this approach by using other data sources as well, such as physiological and behavioral data to assess boredom and medical records to assess health problems. Furthermore, this study focused on boredom in the domain of mathematics and it was conducted using a sample of German secondary school students. As such, it remains to be explored if the present findings generalize to other cultural and school contexts. Limited research points to cultural differences in the experience of boredom between Irish and U.S. citizens [85] or European Canadians and Chinese [86]. In exploring whether our findings also extend to younger age groups, the elementary school version of the AEQ [87] could be used. Further, given the outstanding societal role of the domain of mathematics, we focused our study on this subject domain, expecting that boredom in mathematics would demonstrate substantial links with students’ more general HRQoL. However, the degree to which boredom in other subject domains is linked with HRQoL, too, remains to be explored. Just as the question of whether domain plays a moderating role for student boredom–health outcome links seems to be an intriguing avenue for future research. Finally, we did not take teaching method or parental expectations into account. Instructional design is one of the most reported reasons for boredom [88] and parental expectations can be positively linked with student academic performance but also their depression [89]. Future research could address the role of teaching methods and parental expectations for boredom–HRQoL links.

## 6. Conclusions

In conclusion, the present study provided empirical evidence that boredom is negatively linked with HRQoL and that stronger boredom growth within a semester is linked with lower self-reported health-related quality of life. Teachers, parents, and students should pay attention to boredom as a potential early warning signal for potentially severe, context-specific as well as context-transcending health problems. 

## Figures and Tables

**Figure 1 ijerph-18-06308-f001:**
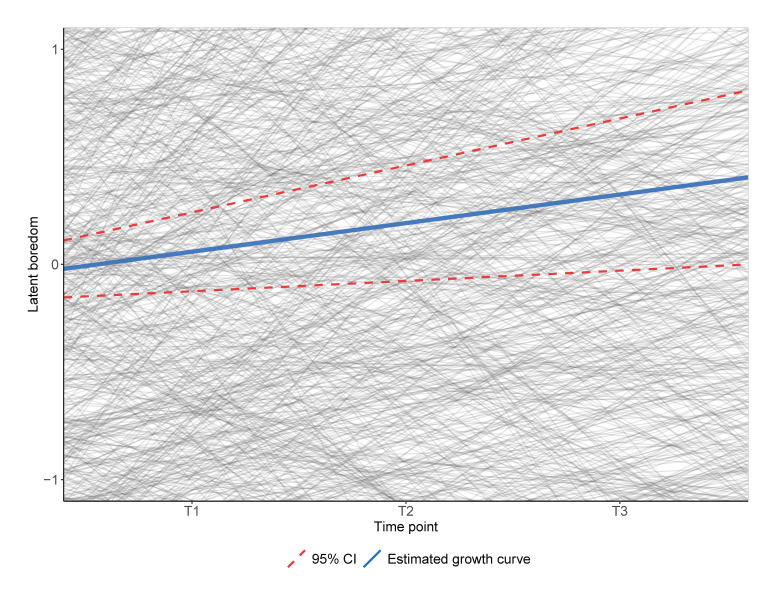
Latent growth trajectory of boredom across the semester. *Y*-axis truncated for an optimal graphical representation of the overall growth trajectory, depicted boredom values extrapolated beyond the observed time points.

**Table 1 ijerph-18-06308-t001:** Latent bivariate correlation coefficients (r) between boredom and HRQoL dimensions.

	Bo T_1_	Bo T_2_	Bo T_3_	PH	PW	PAR	SOC	SCH	GH
Boredom T_1_ (Bo T_1_)	-								
Boredom T_2_ (Bo T2)	0.69	-							
Boredom T_3_ (Bo T3)	0.62	0.72	-						
Physical well-being (PH)	−0.24	−0.18	−0.26	-					
Psychological well-being (PW)	−0.28	−0.28	−0.32	0.55	-				
Autonomy and parent relation (PAR)	−0.18	−0.20	−0.23	0.38	0.52	-			
Social support and peers (SOC)	−0.13	−0.09	−0.10	0.36	0.42	0.40	-		
School environment (SCH)	−0.40	−0.42	−0.49	0.44	0.58	0.48	0.35	-	
General HRQoL (GH)	−0.33	−0.33	−0.39	0.69	0.81	0.69	0.51	0.72	-

*p* < 0.001 for all coefficients with the exceptions of *r*(Bo T1/SOC): *p* = 0.002; *r*(Bo T2/SOC) and *r*(Bo T3/SOC): *p* = 0.013.

**Table 2 ijerph-18-06308-t002:** Latent bivariate correlations between boredom growth parameters and HRQoL dimensions.

	Growth Parameter			
	Intercept		Slope	
	*r*	*p*	*r*	*p*
Physical well-being	−0.24	<0.001	−0.04	0.771
Psychological well-being	−0.28	<0.001	−0.07	0.251
Autonomy and parent relation	−0.18	<0.001	−0.08	0.156
Social support and peers	−0.12	0.005	0.03	0.771
School environment	−0.40	<0.001	−0.15	<0.001
General HRQoL	−0.33	<0.001	−0.11	0.008

## Data Availability

The data presented in this study and the R script for the data analysis are openly available in OSF at https://osf.io/s765q, accessed on 8 June 2021, reference number s765q.

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
