# Peer review of "Boredom Makes Me Sick: Adolescents’ Boredom Trajectories and Their Health-Related Quality of Life"

_ijerph, 2021, doi:10.3390/ijerph18126308_

Round 1

Reviewer 1 Report

The article is well-written and contribute small but neat piece of evidence to the existing corpus. I have only minor comments:

1) Hypotheses should be highlighted in a structural way it should be launched from the new line and to be more easily noticed and followed.

2) I'm not sure why mathematics was chosen for the study (are you hypothesized that mathematics is perceived as important by for instance parents and has higher power to make an influence on children?) Secondly, how can we be sure that boredom related to other subjects does not somehow contribute to lower HRQoL?

3) In the conclusion Authors wrote: "The present study provided empirical evidence that boredom is negatively linked with HRQoL and that stronger boredom growth within a semester is linked with more severe health-related problems" -  in my opinion, it is too simplistic - you have not research kids medically so there are only their self-perception - so the conclusion is rather that latent boredom growth is negatively correlated with self-perception of kids' quality of life/health.

Reviewer 2 Report

The MS is well written and the research is clearly presented

I have only a few minor concerns I list below

Introduction. Maybe they should quote more. For instance I cannot find

Lichtenfeld, S., Pekrun, R., Stupnisky, R. H., Reiss, K., & Murayama, K. (2012). Measuring students' emotions in the early years: the achievement emotions questionnaire-elementary school (AEQ-ES). Learning and Individual differences22(2), 190-201.

Raccanello, D., Brondino, M., Moè, A., Stupnisky, R., & Lichtenfeld, S. (2019). Enjoyment, boredom, anxiety in elementary schools in two domains: Relations with achievement. The Journal of Experimental Education87(3), 449-469.

which are very important papers in the field

Results. If would represent graphically how much the factor 'General HRQoL' affect the growth of boredom

Discussion. A limitation section is lacking. Please add it, for instance how much cultural issues shape the effect, what is the role played by teaching methods and parental exceptations?

Reviewer 3 Report

The issues of boredom and inactivity among adolescents are crucial to characterising the psychosocial functioning of young people. Quantitative research linking the issues of process boredom, quality of life, and prediction of health disorders is one of many attempts to show the key factors determining the quality of life of young people. The study was conducted on a sample of just under 1500 adolescents in Germany. This is an interesting study, however, I have a few questions, comments that may lighten the reviewed study a bit. 

  1. It is useful to add information about the year of the study in the abstract. The issue of the scale of boredom may be crucial in pre-pandemic and pandemic time studies. 
  2. In formulating a theoretical framework, it is useful to present definitions of what boredom is, or how quality of life is understood. These are elementary variables that can be understood in different ways.
  3. As part of the introduction or discussion, it is worth making it clear that boredom can also be triggers - constructive. This does not mean that boredom among adolescents is always a signal of a bad state of life. This should resonate more strongly.
  4. The year of the study in the research procedure was not presented. This is important for the interpretation of the research results. Were the research results collected in the pandemic period?
  5. The year of the study in the research procedure was not presented. This is important for the interpretation of the research results. Were the research results collected in the pandemic period?
  6. Descriptive statistics should be added for variables related to boredom and quality of life.
  7. In addition to correlations, it is worth adding predictive models for the main variables.

    The article is interesting but needs improvement.

Round 2

Reviewer 3 Report

Thank you for the opportunity to read the study again. The authors have very accurately responded to the comments made in the first review. The text in its present form is fully readable for me. I congratulate on the preparation of a valuable scientific study and recommend the text for publication.

Author Response

Thank you for this positive evaluation.